# Estimating and predicting the temporal information of apartment burglaries that possess imprecise time stamps: A comparative study using eight different temporal approximation methods in Vienna, Austria

**Philip Glasner** [ID]<sup>1</sup>, **Michael Leitner** [ID]<sup>2</sup>*, **Lukas Oswald**<sup>3</sup>

**1** Carl Zeiss GmbH, Vienna, Austria, **2** Department of Geography and Anthropology, Louisiana State University, Baton Rouge, Louisiana, United States of America, **3** Infineon Technologies IT-Services GmbH, Villach, Austria

* mleitne@lsu.edu

**Data Availability Statement:** Crime data provided for this study can be requested from the Criminal Intelligence Service Austria (https://www.bundeskriminalamt.at/). The data contain

## Abstract

This research compares and evaluates different approaches to approximate offense times of crimes. It contributes to and extends all previously proposed naïve and aoristic temporal approximation methods and one recent study [1] that showed that the addition of historical crimes with accurately known time stamps to temporal approximation methods can outperform all traditional approximation methods. It is paramount to work with crime data that possess precise temporal information to conduct reliable (spatiotemporal) analysis and modeling. This study contributes to and extends existing studies on temporal analysis. One novel and one relatively new temporal approximation methods are introduced that rely on weighting aoristic scores with historic offenses with exactly known offense times. It is hypothesized that these methods enhance the accuracy of the temporal approximation. In total, eight different methods are evaluated for apartment burglaries in Vienna, Austria, for yearly and seasonal differences. Results show that the one novel and one relatively new method applied in this research outperform all other existing approximation methods to estimate and predict offense times. These two methods are particularly useful for both researchers and practitioners, who often work with temporally imprecise crime data.

## 1. Introduction

Routine activities explain crime patterns and their variations and convergence in space and in time [2,3]. Studies and applications that are based on spatiotemporal crime analysis (see [4] for a review of spatiotemporal methods and several articles published in a recent special journal issue edited by [5]) require data with precise spatial and temporal components. While

potentiality sensitive information and require the data recipient to undergo a background security check.

**Funding:** This research was funded by the Austrian Science Fund (FWF) through the Doctoral College GIScience (DK W 1237-N23) at the University of Salzburg. Co-author Philip Glasner is employed by Carl Zeiss GmbH. Carl Zeiss GmbH provided support in the form of salary for author PG, but did not have any additional role in the study design, data collection and analysis, decision to publish, or preparation of the manuscript. Co-author Lukas Oswald is employed by Infineon Technologies IT-Services GmbH. Infineon Technologies IT-Services GmbH provided support in the form of salary for author LO, but did not have any additional role in the study design, data collection and analysis, decision to publish, or preparation of the manuscript. The specific roles of these authors are articulated in the 'author contributions' section.

**Competing interests:** We have the following interests: Co-author Philip Glasner is employed by Carl Zeiss GmbH. Co-author Lukas Oswald is employed by Infineon Technologies IT-Services GmbH. This does not alter our adherence to PLOS ONE policies on sharing data and materials.

**Abbreviations:** aoristic, Temporal approximation of an imprecisely known offense time using the aoristic method; $aoristic_{ext}$, Temporal approximation of an imprecisely known offense time using the extended aoristic method; RWA, Temporal approximation of an imprecisely known offense time using the retrospectively weighted aoristic method; $RWA_{ext}$, Temporal approximation of an imprecisely known offense time using the extended retrospectively weighted aoristic method; $t_{duration}$, Time duration during which an offense could have occurred, calculated as $t_{end}$–$t_{start}$; $t_{end}$, End time of an (im)precisely known offense time; $t_{mid}$, Mid-point of an imprecisely known offense time; $t_{random}$, Random time within the time range of an imprecisely known offense time; $t_{start}$, Start time of an (im)precisely known offense time.

spatiotemporal crime analysis and research has predominantly focused on the spatial component of crime, unfortunately, the temporal component has been considered in only very few studies. Nevertheless, it is important to have precise temporal information, when investigating and analyzing crime. Therefore, knowing when crime occurs is crucial for forecasting and preventing offenses. For a recently published systematic review on crime forecasting consult [6].

However, for some crime types an exact offense time can be imprecisely known. While occurrence times of crimes against people are often known exactly [7], crime against unguarded property predominantly has a time window reported in which the offense occurred. If crimes were committed without any witnesses, the collection of complete and precise temporal information may be very difficult. Hence, the lack of precisely known times of crime occurrences makes the analysis of crime even more challenging [8]. For these reasons, there is a strong need for methods that estimate crime offense times more accurately. Therefore, law enforcement agencies, as well as researchers, rely on temporal approximation methods in crime analysis. Several temporal approximation methods have recently been discussed in research (e.g., [1,9–12]). The choice of method of offense time approximation has a strong influence on results of (spatio)temporal analyses and a significant impact on law enforcement activities such as deploying resources and solving a particular case.

This paper discusses eight approaches to approximate offense times of apartment burglaries in Vienna, Austria. It builds upon and extends previous research by [1,9,11,12]. The main novel contribution of this paper is the impact that (yearly) seasons have on approximating offense times and how they influence the ability to estimate and forecast apartment burglary offense times. The rationale behind this approach is the mentioning in [2] of seasonal variations in daily routines. This information could be useful for both researchers and practitioners, who deal with temporal crime data and who are tasked to design crime prevention strategies. A second novel contribution of this paper is the introduction of a new technique to more precisely estimate offense times that is based on weighting imprecisely known offense times with exactly known offense times. This is referred to as the *retrospectively weighted aoristic* (RWA) method and is discussed in detail in Subsection 3.3.1 (see below). A second temporal approximation method that is based on the same concept as the RWA method is also applied in this paper. It is called the *retrospectively weighted aoristic*ext ($RWA_{ext}$) method and has previously been proposed and applied by [1], where it is referred to as the *extended retrospective temporal approximation* ($RTA_{ext}$).

## 2. Theoretical background

The routine activity approach has been used to explain temporal variations of crime on different scales, such as the day, the week, the month, or the year (e.g., [13–15]). Certain crime types, such as property crimes (e.g., [16]), experience distinctive seasonal peaks. These seasonal variations in crime are the result of variations in people's routines over the course of the year [2,17]. These seasonal fluctuations of daily routines also have an impact on when crime occurs. This involves seasonal variations of imprecise temporal information of crime data. However, precisely known offense times are an exception but crucial information. As victims may not be home during a (property) offense, the (date and) time of (property) crimes are recorded at the earliest and latest (date and) time they could have occurred. For simplicity, researchers and practitioners are using either the start time (hereafter known as $t_{start}$) or the end time (known as $t_{end}$) of a time range as the actual time in their analyses. However, start and end time only reflect a victim's routine activities, for example, when a burglary victim left and returned to their property [9]. [11] suggested to avoid both start and end methods as approximation for the actual time since they are misleading, even though the start and end time may work well

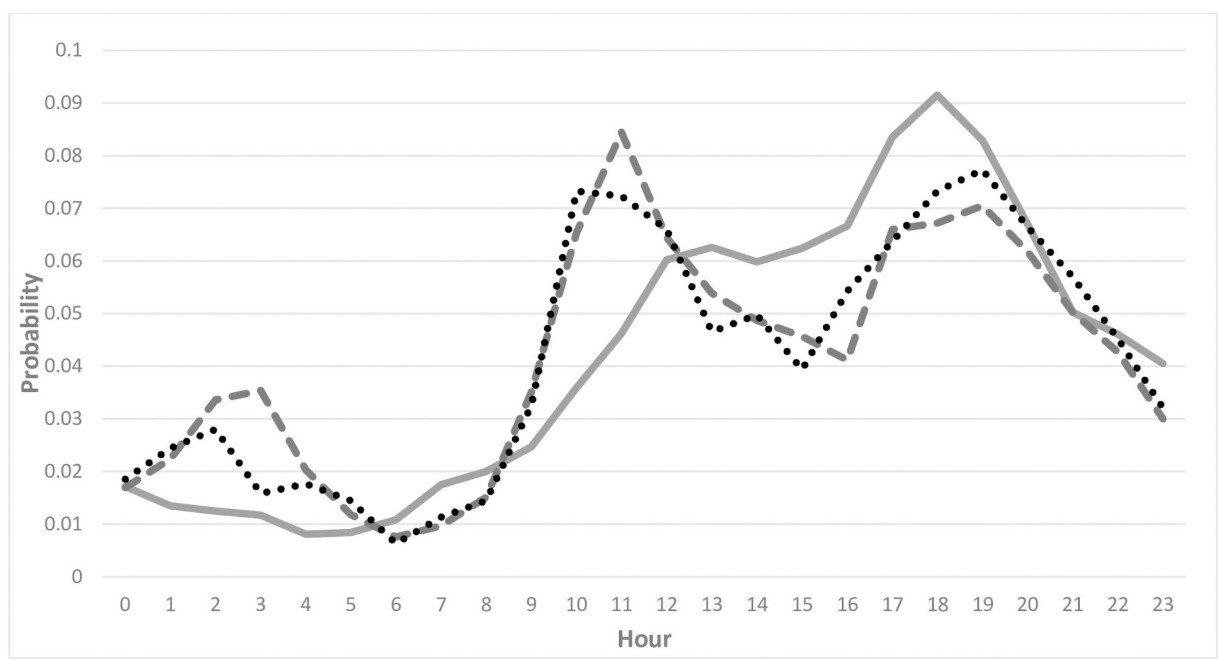

**Fig 1. Hourly distributions of apartment burglaries from 2009–2012 with imprecisely (solid line) and precisely known (dashed line) time stamps; hourly distribution of apartment burglaries from 2013–2015 with precisely known time stamps (dotted line).**

occasionally. Additionally, using either start or end time is not more likely than any other moment within the time range of an offense. This is also true for the point halfway between start and end time, known as $t_{mid}$, assuming that every crime event occurred at the mid-point. A 'random' method, first described by [11], was suggested to minimize this error. Although $t_{random}$ is also not more likely than any other moment within the time range, for a large-enough sample of crimes, $t_{random}$ will be more accurate than $t_{start}$, $t_{mid}$, and $t_{end}$ [11]. As an alternative, [18] suggested to use the aoristic method. In this method, each time unit (e.g., the hour, the weekday, the month, etc.) where a crime could have occurred, receives an equal fraction of 1, whether or not time units are fully covered [19]. If a time window ranges four hours, each of the four individual time units would receive a fraction of 0.25. With crime data of imprecisely known offense times, fractions from each time unit are summed to receive an *aoristic* distribution. To overcome the assignment of equal fractions, even when time units are only partially covered, [12] introduced an extended aoristic method known as *aoristic*$_{ext}$. As an example, consider a crime event that occurred between 17:15 and 20:25. While the aoristic method assigns 0.25 to each of the four hourly units, the *aoristic*$_{ext}$ method weighs each hourly unit by the fraction it covers. In this example, fully covered hours receive an aoristic score of 0.32, while partially covered slots of 45 respectively 25 minutes receive scores of 0.23 and 0.13, respectively. A major advantage of the two aoristic methods compared to the four naïve methods is that probability scores to more than one time slot (e.g., 60 minutes) can be assigned, while each of the four naïve methods assign a score of 1 to only one time slot. This resulted in both aoristic methods to perform significantly better than naïve methods in a study by [12]. However, crimes are not uniformly distributed throughout the day or night, even for shorter time periods, such as a few successive hours (see discussion above and Fig 1 below). This is the main limitation of both aoristic methods, whose main concept is based on the fact that crime occurrences happen uniformly throughout a several hours long time period. Extending aoristic scores with crime occurrences' temporal variability should improve their temporal

approximation and predictive power. This is the approach that is put forward in this research, namely integrating retrospective crimes with exactly known time stamps with both the aoristic and *aoristic*$_{ext}$ methods. The remainder of the paper is organized as follows: Section 3 starts with research objectives and follows with subsections about data preparation and methodology. The latter includes an introduction to the novel temporal approximation method proposed in this paper, as well as statistical tests that are used to evaluate the accuracy of the estimated offense times. Section 4 presents and discusses results of the comparative analysis of estimating offense times of apartment burglaries in Vienna, Austria, based on eight different methods. The final section is the Conclusion.

## 3. Approach

### 3.1. Research objectives

The present study contributes to the existing literature of temporal analysis methods. It is hypothesized that the integration of exactly known offense times enhances the quality of estimating and predicting offense times. For this reason, a new method is proposed that weighs aoristic scores with relative frequencies per time unit (specifically, the hour) of historic crime events, where exact offense times are known. Specifically, this novel method weighs aoristic scores of the aoristic method of imprecisely known offense times with the distribution of exactly known offense times. As mentioned above, it is hereafter referred to as the *retrospectively weighted aoristic* (RWA) method. A second fairly new temporal approximation method that uses the same principles as the RWA method, but takes partially covered time units into consideration (as proposed by [12]) is also applied in this paper. This second approach is from hereon called the *retrospectively weighted aoristic*$_{ext}$ (RWA$_{ext}$) method. It is referred to as the *extended retrospective temporal approximation* (RTA$_{ext}$) in [1], where it has been proposed for the first time. Although, all temporal approximation methods apply retrospective crime data with imprecisely known time stamps, only the RWA and the RWA$_{ext}$ methods use retrospective crime data with exactly known time stamps to weight the distribution of crime data with imprecisely known temporal information. Both methods will be discussed in Subsection 3.3 in more detail. The first research question is whether these two methods are more efficient in estimating and predicting offense times compared to the six methods that have been introduced and evaluated recently. For the comparison of these eight methods, apartment burglaries in Vienna, Austria, are used and statistically evaluated using Spearman's rank correlation coefficient for comparing distributions. Thus, it is hypothesized that Spearman's rank correlation coefficients are higher for the two methods, than for the six existing methods.

Previous research on estimating offense times does not consider the seasonal impact on temporal approximations. While [12] estimated the weekday, the month, or the day in the year for imprecisely known offenses with long time windows, no one has evaluated the impact that seasons have on estimating and predicting offense times. Thus, the entire dataset is divided into a summer, a transition (spring and autumn), and a winter dataset to estimate the impact of seasons on approximating offense times. Again, it is hypothesized that Spearman's rank correlation coefficients are higher for the two RWA and the RWA$_{ext}$ methods, than for the six existing methods.

### 3.2. Data preparation

Vienna, home to approx. 1.85 million people, faces roughly 6,000 reported apartment burglaries every year. Apartment burglaries are a significant problem compared to other European cities [20,21]. The Criminal Intelligence Service Austria, which the federal police in Austria is called, does not analyze residential burglaries together, but differentiates between residential

burglaries to apartments and residential burglaries to houses. In this study, only apartment burglaries are analyzed. All crime data reported in Austria are stored in a database, the so-called Security Monitor [22]. The database is administered by the Criminal Intelligence Service Austria and a dataset of apartment burglaries was made available to the authors for the purposes of this study.

All reported apartment burglary incidents that occurred in Vienna from 2009 to 2015 are analyzed in this research. During this seven-year period, there were a total of 51,387 apartment burglaries in the dataset. In order to have a consistent study design, few criteria have to apply to include an apartment burglary into the study dataset: The event has to be geocoded within the boundaries of Vienna, both the start and end date have to be recorded between 01/01/2009 and 12/31/2015, and the time range must not be larger than 24 hours. This is compliant with [23], who recommended to remove these crimes when approximating offense times. [19] added that aoristic analyses are complicated for events with time ranges larger than 24 hours. Three events were not correctly geocoded and are outside of the study area. For 192 apartment burglaries, the date of $t_{start}$ was reported before 2009. Finally, 13,913 apartment burglaries have a reported time range of more than 24 hours. While two or three criteria may apply to one single event, in total 13,940 (27.1%) events are excluded from the dataset and not included in the analysis.

For the following analysis, three datasets are extracted from the remaining dataset of 37,447 apartment burglaries. The first dataset contains apartment burglaries with imprecisely known offense times that were recorded in the years 2009 to 2012. In this research, imprecisely known offense times are defined as offenses that occurred within a 24-hour period and with the offense duration ($t_{duration}$), calculated as $t_{end}$—$t_{start}$, being larger than 0. Based on the first dataset, temporal approximations are made with each of the eight methods. The second dataset contains apartment burglaries with the exact time of occurrence to be known for the same four years as the first dataset (2009 to 2012). In this research, the exact time of occurrence is defined as an offense time that occurred within a 24-hour period and has an offense duration equal to 0. This dataset is used for weighting the aoristic scores when applying either the RWA or the $RWA_{ext}$ method. Using these two datasets, predictions of offense times for the years 2013 to 2015 are made. To evaluate these predictions, the third dataset uses apartment burglaries from 2013 to 2015 with exactly known occurrence times (i.e., the offense duration is equal to 0). The dataset of precisely known offense times is queried from the total dataset, where the start time equals the end time. As [24] review in their project report about residential burglaries in Austria, these exactly known time stamps are the result of, for example, observations by victims or neighbors, burglar alarms, or CCTV footages.

For this reason, the question may arise whether the sample of offenses with precisely known time stamps (see Table 1) is representative of the population of crime offenses or, at least, not to be systematically different from offenses with imprecisely known time stamps. [11] argue that it is unlikely that a set of crime data represents a random sample of the crime population. However, given the dataset of reported crimes and crimes with precisely known

**Table 1. Seasonal structure of the study dataset of apartment burglaries in Vienna, 2009–2015.**

|  | Imprecisely known time stamp | Precisely known time stamp | Precisely known time stamp |
|---|---|---|---|
|  | (2009–2012) | (2009–2012) | (2013–2015) |
| Summer | 3 875 | 686 | 491 |
| Transition | 9 798 | 1 620 | 1 109 |
| Winter | 5 887 | 998 | 612 |
| Total | 19 560 | 3 304 | 2 212 |

time stamps, it is hypothesized that there is a strong association between crimes with precisely known and imprecisely known time stamps from 2009 to 2012. A strong association would indicate that the sample of offenses with exactly known time stamps is representative of the entire dataset. Furthermore, it is hypothesized that there is a strong correlation between offenses with precisely known occurrence times for 2009 to 2012 and for 2013 to 2015. Furthermore, a strong correlation would indicate that the principle to predict, when crimes are more likely to occur, is justifiable. These hypotheses will be tested at the beginning of Section 4.

Fig 1 shows the distributions of these three datasets of offenses with precisely and imprecisely known offense times. It may be assumed that offenses that occur in the evening are more likely to have an exactly known time, because more people in residential areas may witness offenses, when they return home from work. However, offenses with precisely known time stamps show a peak around noon and in the evening times. The latter observation was also made by [24] and [25], who concluded that residential burglaries mostly occur in the winter months from November to March, when dusk comes early in Austria. Before the residents return home from work, burglars still have a short time period to commit a crime, when it is already dark outside and properties are unattended and unsecured. Given the large amount and potential to prevent these burglaries, these burglaries that occur during winter evenings are widely known as "dusk burglaries" in Austria.

To account for seasonal variations, all three datasets are further subdivided into a summer, a transition, and a winter dataset. The summer dataset represents events that occurred in the months of June, July, and August; the transition dataset contains events from March to May and from September to November; and the winter dataset contains events from the months December, January, and February. This classification of months to seasons is a common practice at the Criminal Intelligence Service Austria and reflects seasonal variations in sunrise and sunset. This classification is also in line with [16], who experienced distinctive seasonal peaks within property crimes such as residential burglaries. Table 1 shows the counts of apartment burglary events for each of the nine datasets that are used in this study. These nine datasets sum up to 25,076 apartment burglaries. The difference between these 25,076 apartment burglaries and the 37,447 apartment burglaries reported above (i.e., 12,371 burglaries) are the apartment burglaries from 2013 to 2015 with imprecisely known time stamps. These burglaries are not used in the analysis.

### 3.3. Methodology

**3.3.1. Temporal approximation methods.**   In the analysis of this study eight methods for temporal approximation of imprecisely known offense times are evaluated. The first four are methods where one single moment within the temporal range of an unknown offense time is used as the approximation method. This moment receives a weight of 1 in each method, respectively. Using an example of a crime event with an imprecisely known time that occurred between 17:15 and 20:25, the approximation time is $t_{start}$ = 17:15 for the start time method, $t_{end}$ = 20:45 for the end time method, and $t_{mid}$ = 18:50 for the mean time method. For the random method, a random time within the time range, for example, $t_{random}$ = 19:24, is used as approximation for an imprecisely known offense time.

The remaining four methods are based on the concept of aoristic weights of imprecisely known offense times. The aoristic method [19] equally assigns each time unit within the time window of an offense an equal weight that sums up to 1. For example, if an event occurs between 17:15 and 20:25, for each (or parts) of the four hours, an equal weight of 0.25 is assigned as aoristic score (Fig 2a). This process is then repeated for each event, where the

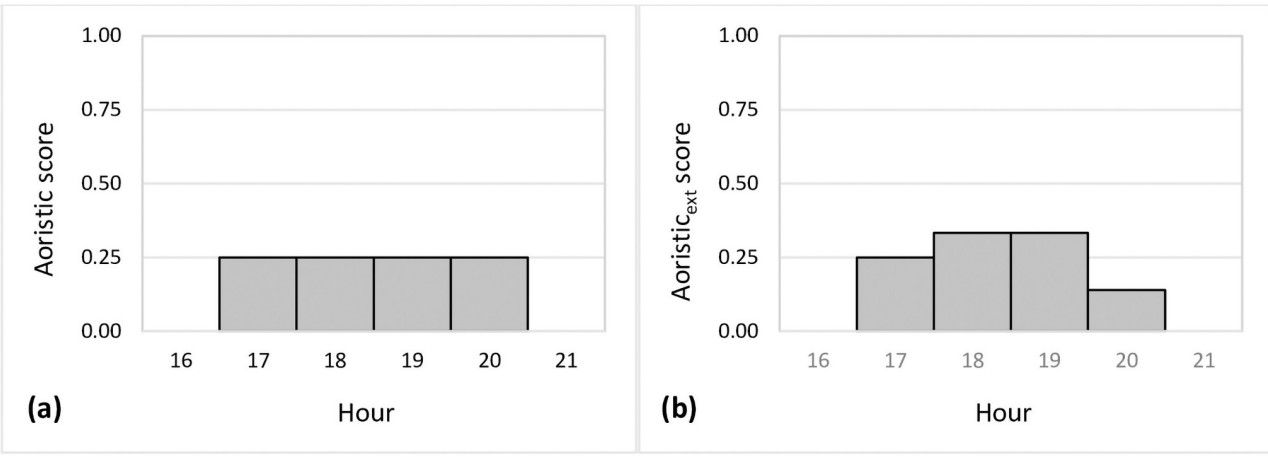

**Fig 2. Distribution of scores of the aoristic (a) and aoristic$_{ext}$ (b) methods to estimate offense times.**

offense time is imprecisely known and results in accumulated aoristic scores in each time unit [19]. In this research, the 24 hours of a full day are used as time units.

The aoristic$_{ext}$ method, proposed by [12], extends the aoristic method by assigning proportional weights to time units that are only partially covered. In the previous example with an event that occurred between 17:15 and 20:25, the time window spans 190 minutes. Each of the hourly fractions (of 45, 60, 60, and 25 minutes) are then divided by the length of the time span. The resulting aoristic scores across the hours are again summing up to 1 (Fig 2b). When comparing Fig 2a with Fig 2b, it is recognizable that imprecisely known times of occurrences with partially covered hours have a significant impact on the distribution of aoristic$_{ext}$ scores, because more detailed temporal information is available.

The novel RWA, and the relatively novel RWA$_{ext}$ methods are weighing the aoristic respectively the aoristic$_{ext}$ scores with relative frequencies per hour of historic crime events, where exact offense times are known. It is hypothesized that the integration of precisely known offenses enhances the quality of the temporal approximation. It is also assumed that the more precise crime data that are available for weighting aoristic scores, the better the results are. In other words, a small number of events with precisely known offense times will more likely bias the temporal approximation.

Referring to the previous example, let us consider a (fictive) distribution of (precisely) known crime events with relative frequencies in hourly units: Unit$_{17}$ (17:00 to 17:59): 5.6%, unit$_{18}$ (18:00 to 18:59): 7.9%, unit$_{19}$ (19:00 to 19:59): 11.2%, and unit$_{20}$ (20:00 to 20:59): 9.7%. The relative frequencies sum up to 100% for crime events distributing to each of the 24 hours. These relative frequencies are used as weights and are multiplied with aoristic or aoristic$_{ext}$ scores. The resulting scores are: Score$_{17}$ = 0.25 x 0.056 = 0.014, score$_{18}$ = 0.25 x 0.079 = 0.020, score$_{19}$ = 0.25 x 0.112 = 0.028, score$_{20}$ = 0.25 x 0.097 = 0.024. Since these scores do not sum up to 1, they are standardized by the sum of the scores in order to receive the RWA scores. In this example, the sum of the individual scores is 0.086, resulting in the following RWA scores: RWA score$_{17}$ = 0.014/0.086 = 0.163, RWA score$_{18}$ = 0.020/0.086 = 0.230, RWA score$_{19}$ = 0.028/0.086 = 0.326, RWA score$_{20}$ = 0.024/0.086 = 0.282. Summing up these individual RWA scores results in a value of 1 and make them comparable to other temporal approximation methods. Eq (1) shows the principle of the RWA method, where an hourly *aoristic score$_h$* is multiplied with the relative frequency of precisely known crime events for the same

corresponding hour.

$$RWA\ score_h\ =\ \frac{aoristic\ score_h * known\ crimes_h}{\sum(aoristic\ score_h * known\ crimes_h)} \tag{1}$$

The concept of calculating $RWA_{ext}$ scores is the same as for RWA scores with the exception that relative frequencies are weighted with $aoristic_{ext}$ scores. This means that the $RWA_{ext}$ method considers whether time units are fully or partially covered. In the following (fictive) example, the same timeframe (17:00 to 20:59) and relative frequencies in hourly units of 5.6% (17:00 to 17:59), 7.9% (18:00 to 18:59), 11.2% (19:00 to 19:59), and 9.7% (20:00 to 20:59) from historic crime events with exactly known offense times, as for the calculation of the RWA scores, are applied in the calculation of the $RWA_{ext}$ scores. These relative frequencies are used as weights and multiplied with $aoristic_{ext}$ scores, resulting in: $Score_{17}$ = 0.056 x 0.24 = 0.013, $score_{18}$ = 0.079 x 0.32 = 0.025, $score_{19}$ = 0.112 x 0.32 = 0.036, $score_{20}$ = 0.097 x 0.13 = 0.013. Since these scores do not sum up to 1, they are standardized by the sum of the scores in order to calculate $RWA_{ext}$ scores. The sum of the individual scores is 0.087 (due to rounding, slightly higher than 0.086, the sum of the values used for calculating the RWA scores) resulting in the following $RWA_{ext}$ scores: $RWA_{ext}$ $score_{17}$ = 0.013/0.087 = 0.149, $RWA_{ext}$ $score_{18}$ = 0.025/ 0.087 = 0.287, $RWA_{ext}$ $score_{19}$ = 0.036/0.087 = 0.414, $RWA_{ext}$ $score_{20}$ = 0.013/0.087 = 0.149. Summing up all four individual $RWA_{ext}$ scores results in a value of 1, which make the scores comparable to the other seven temporal approximation methods. Eq (2) shows the principle of the $RWA_{ext}$ method, where an hourly $aoristic_{ext}$ $score_h$ is multiplied with the relative frequency of exactly known crime events for the same corresponding hour.

$$RWA_{ext}\ score_h\ =\ \frac{aoristic_{ext}\ score_h * known\ crimes_h}{\sum(aoristic_{ext}\ score_h * known\ crimes_h)} \tag{2}$$

Fig 3a and 3b show the RWA and $RWA_{ext}$ scores for the same time period of the fictive distribution. Comparing the distribution of the scores, it is recognizable that both distributions experience a distinctive peak from 19:00 to 19:59, resulting from a large relative frequency of crime events with exactly known offense times in this time unit.

**3.3.2. Statistical evaluation.** The study design uses imprecisely and precisely known time stamps for apartment burglaries in Vienna, Austria, from 2009 to 2012 to forecast offense times for the years 2013 to 2015. These forecasted scores are evaluated with apartment

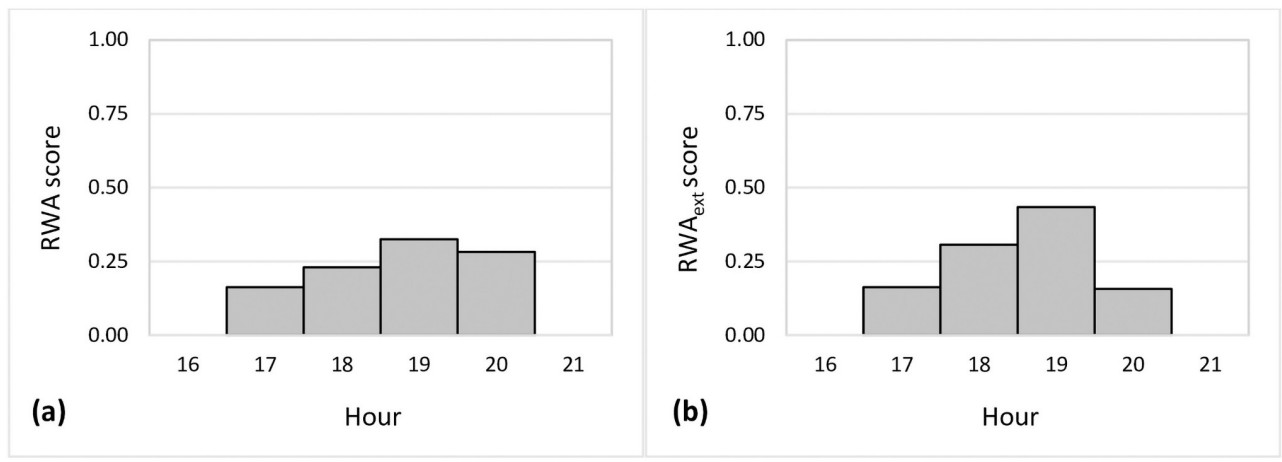

**Fig 3. Distribution of scores of the RWA and $RWA_{ext}$ methods to estimate offense times.**

burglaries with exactly known offense times from the same years from 2013 to 2015. To compare and evaluate the relationship between distributions of hourly scores derived from eight approximation methods tested in this research with the evaluation dataset of exactly known offense times, the Spearman's rank correlation coefficient (also known as Spearman's $\rho$) is used. This test is a non-parametric rank-based correlation test that measures the strength of the association between two datasets. Values of Spearman's $\rho$ range from -1 to 1, where 0 means no correlation, -1 to $<$0 a negative correlation and $>$0 to 1 a positive correlation. Spearman's $\rho$ has been favored over Pearson's correlation coefficient, which requires normally distributed datasets, and Kolmogorov-Smirnov, a non-parametric test. None of the datasets were found to be normally distributed. Kolmogorov-Smirnov, however, is not applicable, since the data used in this study are binned [26].

Eight Spearman's coefficients are the result of the correlation analysis of the eight approximation methods and the evaluation dataset of offenses with exactly known time stamps from 2013 to 2015. In order to statistically compare these correlation coefficients, values of Spearman's $\rho$ are transformed using the Fisher-z transformation. These transformations are used, because the comparison of non-standardized correlation coefficients is difficult to work with. However, distributions of z-values are approximately normally distributed, which make them suitable for a comparison. Then, a test statistic is calculated to determine, if there is a statistically significant difference between two z-values of correlation coefficients. A p-value is derived under the null hypothesis of equal correlation [27,28].

Fig 4 shows the analysis design of this study, with each temporal approximation method being evaluated with crimes possessing exactly known time stamps using Spearman's $\rho$. The RWA and RWA$_{ext}$ methods additionally use crimes with precisely known temporal information to retrospectively weight aoristic scores of crimes with imprecisely known time stamps.

## 4. Results and discussion

In this section results of applying the eight methods for temporal approximation to apartment burglaries with imprecisely known offense times in Vienna, Austria, are shown. First, in Subsection 4.1., it will be analyzed if distributions of offenses with precisely and imprecisely known time stamps show a strong association and are therefore a representative sample to weigh offenses with imprecisely known time stamps. In Subsection 4.2., the entire dataset is used to forecast offense times and in Subsection 4.3., seasonal differences of estimating offense times are discussed. Finally, Subsection 4.4. shows the limitations that this study has.

### 4.1. Representativeness of the sample of apartment burglaries with precisely known time stamps

First, the hypothesis that apartment burglaries with imprecisely and precisely known offense times from 2009 to 2012 are from the same distribution will be analyzed. A correlation analysis using Spearman's rank correlation coefficient results in a Spearman's $\rho$ of 0.797 and is statistically significant at p = 0.001. This means that the distribution of the sample of offenses with precisely known time stamps from 2009 to 2012 (3,304 events, see Table 1) is similar to the distribution of offenses with imprecisely known time stamps from 2009 to 2012 (19,560 events). This, however, does not fully answer whether this sample is representative of the total population of apartment burglaries. But given the nature of the occurrence of apartment burglaries, this limitation is known and cannot be solved. Therefore, the sample of offenses with exactly known time stamps is assumed to be representative of the total population of apartment burglaries and suitable for the analysis of this study.

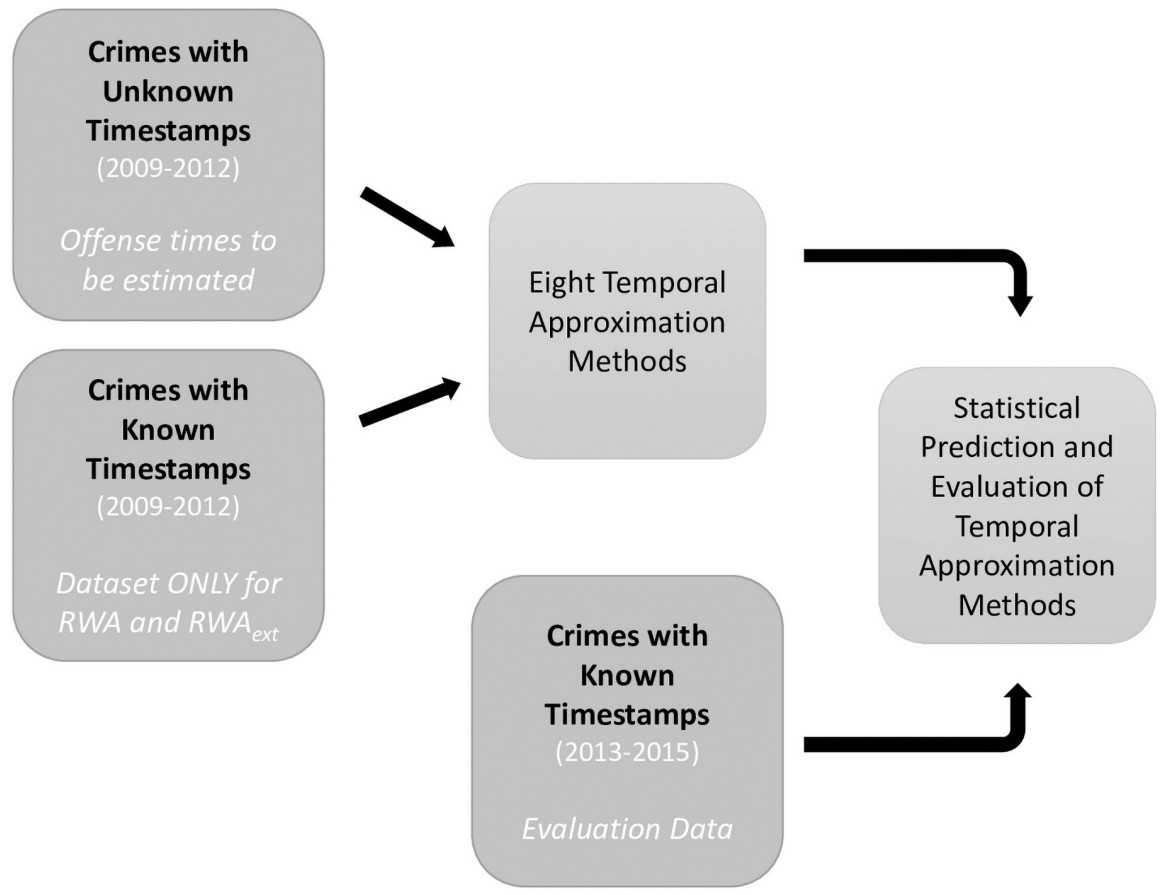

**Fig 4. Workflow to apply and evaluate temporal approximation methods.**

Additionally, offenses with exactly known time stamps from 2013 to 2015 also show a strong correlation to offenses with exactly known time stamps from 2009 to 2012 according to a Spearman's $\rho$ of 0.954 (p = 0.001). This means that the distribution of offenses with precisely known time stamps (from 2013 to 2015) that are used to evaluate the approximated offense times is very similar to the distribution of offenses with precisely known time stamps that are used to weigh the offenses with imprecisely known time stamps (both from 2009 to 2012; see Fig 4 for the analysis workflow).

## 4.2. Forecasting and evaluating offense times using eight different temporal approximation methods

In this analysis, eight approximation methods applied to temporal imprecisely known apartment burglaries from 2009 to 2012 are used to forecast offense times for the years 2013 to 2015. The approximated time stamps for apartment burglaries are evaluated with apartment burglaries from 2013 to 2015 that possess exactly known offense times. Table 2 shows results of the Spearman's $\rho$ comparing forecasted and approximated distributions of apartment burglaries with imprecisely known offense times from 2009 to 2012 to apartment burglaries with precisely known offense times from 2013 to 2015. As hypothesized, Spearman's $\rho$ shows highest correlation coefficients for both RWA and RWA$_{ext}$ methods. These results are different from [12], who concluded that the aoristic method was the most suitable method for

**Table 2. Statistical results comparing eight approximated offense times with exactly measured offense times.**

|  | Spearman's $\rho$ |
| --- | --- |
| $t_{start}$ | 0.255 |
| $t_{end}$ | 0.781* |
| $t_{mid}$ | 0.755* |
| $t_{random}$ | 0.692* |
| aoristic | 0.691* |
| aoristic$_{ext}$ | 0.698* |
| RWA | 0.861* |
| RWA$_{ext}$ | 0.846* |

* Statistically significant (p = 0.001).

estimating offense times for residential burglaries in Sweden. However, in our study, the aoristic method was outperformed by RWA, RWA$_{ext}$, $t_{end}$, and $t_{mid}$. However, $t_{end}$ and $t_{mid}$ should be used with caution (or even avoided as suggested by [11]), because using the end or mid time methods is not more likely than any other moment within the time range. In this study, the worst temporal approximation method is $t_{start}$. All correlation coefficients but the Spearman's $\rho$ for $t_{start}$ are statistically significant at p = 0.001. The pair-wise comparisons of the Spearman's *coefficients* revealed that the differences between each method and $t_{start}$ are statistically significant (p<0.05) but none of the other pair-wise comparisons are statistically significant.

## 4.3. Forecasting and evaluating offense times using eight different temporal approximation methods for a summer, a transition, and a winter season

Seasons of a year indicate a significant impact on apartment burglaries in Vienna, Austria. Peaks in apartment burglaries are highest around noon in summer, but around evening hours in winter. These seasonal fluctuations and their impact on estimating offense times are evaluated by dividing the dataset analyzed in Subsection 4.1. into three seasons, namely summer (June to August), a transition season (March to May and September to November), and winter (December to February). Table 3 shows results of the Spearman's $\rho$ comparing forecasted and approximated distributions of apartment burglaries with imprecisely known offense times from 2009 to 2012 to apartment burglaries with exactly known offense times from 2013 to 2015 for the three seasons.

During summer, the RWA method performs best using the Spearman's $\rho$, closely followed by RWA$_{ext}$. Interestingly, nearly all methods (except $t_{start}$) outperform $t_{end}$, which was within the leading methods, when the entire year was analyzed (Subsection 4.1.). For the transition period (spring and fall), results are similar to the results for the summer with RWA and RWA$_{ext}$ in addition to correlation coefficients. During winter, when the peak in offense times of apartment burglaries with exact time stamps is in the late afternoon and evening, the end method $t_{end}$ shows the highest Spearman's $\rho$, closely followed by RWA$_{ext}$ and RWA. A possible explanation is that the shift of the peak of apartment burglaries to the evening hours during the winter season coincides with daily routines of property owners, who return to their property at about the same time. Therefore, the reported end time seems to be a good approximation for the actual offense time. But, however, as this phenomenon can only be seen during the winter season, $t_{end}$ (as well as any other method that uses one single moment within the time range) should be used with caution. Pair-wise comparisons of Spearman's $\rho$ revealed that in

**Table 3. Statistical results of comparing eight approximated offense times with exactly measured offense times by seasons.**

| | Spearman's $\rho$ | | |
| | S | T | W |
|---|---|---|---|
| $t_{\text{start}}$ | 0.131 | 0.220 | 0.233 |
| $t_{\text{end}}$ | 0.418* | 0.701** | 0.874** |
| $t_{\text{mid}}$ | 0.721** | 0.726** | 0.710** |
| $t_{\text{random}}$ | 0.609** | 0.653** | 0.707** |
| aoristic | 0.620** | 0.650** | 0.746** |
| aoristic$_{\text{ext}}$ | 0.651** | 0.650** | 0.753** |
| RWA | 0.813** | 0.790** | 0.853** |
| RWA$_{\text{ext}}$ | 0.805** | 0.789** | 0.854** |

S . . . summer (Jun-Aug); T . . . transition (Mar-May & Sep-Nov); W . . . winter (Dec-Feb).

* Statistically significant (p<0.05).

** Statistically significant (p = 0.001).

the summer season all methods but $t_{\text{end}}$ perform significantly better than $t_{\text{start}}$ (p<0.05), and both RWA and RWA$_{\text{ext}}$ perform significantly better than $t_{\text{end}}$ (p<0.05). All other pair-wise combinations are not statistically significant. In the transition and winter season the differences of Spearman's $\rho$ between each method and $t_{\text{start}}$ are statistically significant (p<0.05) but all other pair-wise comparisons are not statistically significant.

## 4.4. Limitations

This study shows that the novel RWA and the relatively new RWA$_{\text{ext}}$ approximation methods clearly outperform other methods for temporal approximation. However, both methods require a considerable number of crime data with precisely known (retrospective) offense times to weigh both aoristic scores. The most important limitation of offenses with exactly known time stamps being a representative sample of the total population has been discussed and unfortunately, there is no way to overcome it. However, it was analyzed that offenses with both precisely and imprecisely known time stamps correlate to a high degree. This means that both distributions are similar to each other. Additionally, offenses with exactly known time stamps from 2009 to 2012 also show a strong correlation to offenses with exactly known time stamps from 2013 to 2015 with a highly statistically significant Spearman's $\rho$ of 0.954. This strong correlation prompted [1] to develop a novel temporal approximation method, referred to as the retrospective temporal analysis (RTA) method. This method uses solely historical crimes (in the case of [1] burglary crimes are applied) with accurately known time stamps to approximate time stamps of future crimes for which accurate temporal information is not available. Results showed that the RTA method is at least equal or clearly superior to all other temporal approximation methods tested and evaluated in [1].

Based on the above results in this research, the RWA method seems to be slightly (but not statistically significantly) more suitable than RWA$_{\text{ext}}$ method. [12] argue that this is likely due to the imprecise report of offense times. Mostly, crime events are reported with times rounded to full hours, and therefore, both the aoristic$_{\text{ext}}$ and RWA$_{\text{ext}}$ methods do not show better results than the aoristic or the RWA method, respectively. This observation can be confirmed with a brief analysis shown in Table 4. In total, 57.5% of apartment burglaries of the entire dataset used in this research (51,387 apartment burglaries from 2009 to 2015) start at any full hour, while 38.8% of all crime events are reported to end at any full hour. The end time is likely to be

**Table 4.  Start and end times (by minutes) of apartment burglaries of the entire dataset of apartment burglaries in Vienna, 2009–2015.**

|  | $t_{start}$ | | $t_{end}$ | |
| --- | --- | --- | --- | --- |
|  | **Count** | **%** | **Count** | **%** |
| :00 | 29 569 | 57.5 | 19 959 | 38.8 |
| :15 | 2 667 | 5.2 | 3 927 | 7.6 |
| :30 | 10 901 | 21.2 | 12 036 | 23.4 |
| :45 | 2 653 | 5.2 | 4 082 | 7.9 |
| In-between | 5 597 | 10.9 | 11 383 | 22.2 |
| Total | 51 387 | 100.0 | 51 387 | 100.0 |

more precise, because victims usually know better, when they returned to their property but do less likely remember, at what exact time they left.

In this research, we have not tested other time units (other than hours over a 24-hour time period) to estimate, for example, the weekday or the month. Additionally, we have not tested shorter time units than the hour, for example, 15 or 30 minutes, to estimate offense times. Based on the results in Table 4, using time units of 15 or 30 minutes are not expected to show more precise results in estimating offense times. Hence, small units are likely to produce distributions that are spikey and poorly fitted to the actual distribution of exactly known offenses [29].

## 5. Conclusion

Many applications in crime analysis require precise temporal information. However, for a considerable amount of crime events, especially property crimes, only imprecise temporal information is available. Therefore, there is an urgent need to further develop and evaluate temporal approximation methods in research and in practice to alleviate this problem. In this study, six existing, one relatively new, and one novel temporal approximation methods were evaluated regarding their effectiveness to more precisely estimate and predict offense times.

Results show that the novel RWA and the relatively new RWA$_{ext}$ methods that include retrospective data with exactly known offense times, slightly outperform earlier developed methods, such as the aoristic or the aoristic$_{ext}$ method. Differences between the results of estimating offense times using the RWA and RWA$_{ext}$ methods that were discovered in this research can be based on imprecise reporting of crime offense times. Additional results show that estimating offense times is significantly influenced by seasons of the year. This implies that regions with other seasonal patterns than the study area in this research may have a different impact on daily routines and offense times. It should be noted that this research only uses apartment burglaries in the city of Vienna, Austria. Therefore, results and the most suitable method to use for a temporal approximation may vary for other crime types, as well as for other study areas.

However, results from this research already confirm previous research [1] that the addition of historical crimes with accurately known time stamps to temporal approximation methods can outperform traditional approximation methods, including all naïve and aoristic temporal approximation methods. These results are consistent across different crime types, including apartment, car, and house burglaries in [1] and again apartment burglaries in this research, and also across seasons of the year in case of apartment burglaries (this research). While these new approximation methods have not yet been applied to all potential crime types for which time stamps are traditionally inaccurate, such as car thefts or pickpocketing, we believe that

results from this research can be generalized to these crime types, as well. In general, we are also confident that temporal approximation methods that are enhanced with accurately known time stamps from the recent past, would also perform superior to traditional approximation methods across study areas outside of the study sites discussed in this research and in Oswald and Leitner (2020). Of course, our believes can only be verified with the replication of similar studies across many different study sites.

For these reasons, we are interested to evaluate the RWA and the $RWA_{ext}$ methods for other crime types and/or for other study areas in the near future. Additionally, we are interested to explore whether there are "inter-geographical" differences in estimating offense times, for example, to evaluate whether different patterns to estimate offense times exist within one and the same study area. We are thinking of weighting offenses both spatially and temporally with "near" crime events with exactly known offense times. This idea is inherited from the concept of near repeat victimization (for example see [30]), where the risk of another event to occur decreases with space and time. Due to small sample sizes that are expected to be used in this approach, we reference [12], who confirmed that small sample sizes are still able to effectively estimate offense times.

## Author Contributions

**Conceptualization:** Philip Glasner.

**Data curation:** Philip Glasner.

**Formal analysis:** Philip Glasner, Lukas Oswald.

**Funding acquisition:** Michael Leitner.

**Investigation:** Philip Glasner.

**Methodology:** Philip Glasner, Lukas Oswald.

**Project administration:** Michael Leitner.

**Supervision:** Michael Leitner.

**Writing – original draft:** Philip Glasner, Lukas Oswald.

**Writing – review & editing:** Michael Leitner.

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
