## [Decision Letter · Decision Letter 0]

23 Dec 2020

PONE-D-20-32940

Estimating and predicting the temporal information of apartment burglaries that possess imprecise timestamps: A comparative study using eight different temporal approximation methods in Vienna, Austria

PLOS ONE

Dear Dr. Leitner,

Thank you for submitting your manuscript to PLOS ONE. After careful consideration, we feel that it has merit but does not fully meet PLOS ONE’s publication criteria as it currently stands. Therefore, we invite you to submit a revised version of the manuscript that addresses the points raised during the review process.

We look forward to receiving your revised manuscript.

Kind regards,

Chaowei Yang

Academic Editor

PLOS ONE

Journal Requirements:

We note that one or more of the authors are employed by a commercial company: ZEISS, Infineon Technologies IT.

2.1. Please provide an amended Funding Statement declaring this commercial affiliation, as well as a statement regarding the Role of Funders in your study. If the funding organization did not play a role in the study design, data collection and analysis, decision to publish, or preparation of the manuscript and only provided financial support in the form of authors' salaries and/or research materials, please review your statements relating to the author contributions, and ensure you have specifically and accurately indicated the role(s) that these authors had in your study. You can update author roles in the Author Contributions section of the online submission form.

2.2. Please also provide an updated Competing Interests Statement declaring this commercial affiliation along with any other relevant declarations relating to employment, consultancy, patents, products in development, or marketed products, etc.  

Please respond by return email with an updated Funding Statement and Competing Interests Statement and we will change the online submission form on your behalf.

Reviewers' comments:

Reviewer's Responses to Questions

**Comments to the Author**

1. Is the manuscript technically sound, and do the data support the conclusions?

Reviewer #1: Partly

Reviewer #2: Yes

2. Has the statistical analysis been performed appropriately and rigorously? 

Reviewer #1: Yes

Reviewer #2: Yes

3. Have the authors made all data underlying the findings in their manuscript fully available?

Reviewer #1: Yes

Reviewer #2: Yes

4. Is the manuscript presented in an intelligible fashion and written in standard English?

Reviewer #1: Yes

Reviewer #2: Yes

5. Review Comments to the Author

Reviewer #1: The manuscript “Estimating and predicting the temporal information of apartment burglaries that possess imprecise timestamps: A comparative study using eight different temporal approximation methods in Vienna, Austria” introduced two novel temporal approximation methods and evaluated apartment burglaries in Vienna, Austria, for yearly and seasonal differences. The manuscript is a methodological study with an element of a case study. The analysis the paper promises is good, but the analysis delivered is less exciting. The methods are remarkably similar to existing studies. Therefore, I suggest emphasizing and in-depth analysis of seasonal differences.

1. Both the RWA, RWAext novel methods are similar to the methods provided in another published article (doi:10.3390/ijgi9060386). Assumptions are also similar. So the methods are not very novelty. The difference between the novel methods in this paper and the existing methods should be identified.

2. Line 42: Please also include other keywords to mention the temporal approximation methods you introduced.

3. Line 269: As the Euqation (1) you mentioned shows the principle of the RWA method, please also use an Equation to show the principle of the RWAext method.

4. Line 278: It seems that the number of Section “3.3.1 Statistical Evaluation” was wrong.

5. Line 336: It will be better to additionally compare approximated distributions of apartment burglaries with unknown times from 2013 to 2015 to apartment burglaries with known offense times from 2013 to 2015. I leave the final decision to the authors.

6. Line 450: Authors should add other similar work to improve the scientific strength of the work. There are no references published in the past two years in the references section.

7. Line 519: Incomplete legend has been provided in Figure 1. And the meaning of vertical axis is not clear.

Reviewer #2: This study addresses the question of how to handle temporal uncertainty in the time coding of crime incidents. It compares 8 different strategies for addressing such uncertainty, and compares the results. It uses data from Vienna for these comparisons.

I like the general idea of the research question, as it is certainly an important one. It seems like the data are useful for addressing the question, although there were a few details I was unclear on.

It is useful to compare strategies to the two aoristic strategies that are included, given that they are often used in studies. I do think these are quite naïve strategies, particularly the initial aortic one that does not account for partial hours, so it might be useful to at least point out a few of the potential limitations we would expect from these strategies based on their assumptions.

I found myself somewhat confused about what exactly the data provides. Is there different time information for crime incidents in different years? At points, it seemed like there was, and that fed into the strategy used. But at other points it seemed like there was not. It would be good to add a paragraph that makes it explicit what temporal information is available in which years. And which years have the “gold standard” of correct time, and how exactly is that “gold standard” determined? This should be presented in a very clear fashion, as currently the information about this seems to be spread at various points in the data section, and it was not very clear at points. This may mostly be a re-ordering of information in the data section.

It was not clear to me at points whether the author(s) were referring to incidents with exact time of incident, incidents with a range of time period, or incidents with no time information (I’m not sure how the latter would occur; it seems like this should just have a very wide time range). But making this clearer would help the readability at points.

I wondered if another strategy would be to use time from the beginning of the unknown time period? If a house was being cased, it might be that it would be burglarized shortly after the people left, regardless of what time that is. Just a thought.

On page 15, it was not clear to me whether the correlation reported is based on specific cases, or the distribution? I believe it’s the latter, but that should be clarified.

These are some interesting findings. It would be useful to talk about the potential generalizability of them. There are many possible issues. How generalizable are the temporal patterns of crime in this city? How generalizable is this particular crime type to others? It would be good to explicitly talk about these issues in the discussion section, as they would be important to potential users of the technique, especially if they did not have the type of data used here to calibrate the model. They would need to assume some of the patterns in the crime data here, so how reasonable would that be?

6. PLOS authors have the option to publish the peer review history of their article (what does this mean?). If published, this will include your full peer review and any attached files.

Reviewer #1: No

Reviewer #2: No

---

## [Author Response · Author response to Decision Letter 0]

20 May 2021

Reviewers' comments:

Reviewer's Responses to Questions

Comments to the Author

1. Is the manuscript technically sound, and do the data support the conclusions?

Reviewer #1: Partly

Reviewer #2: Yes

2. Has the statistical analysis been performed appropriately and rigorously? 

Reviewer #1: Yes

Reviewer #2: Yes

3. Have the authors made all data underlying the findings in their manuscript fully available?

Reviewer #1: Yes

Reviewer #2: Yes

4. Is the manuscript presented in an intelligible fashion and written in standard English?

Reviewer #1: Yes

Reviewer #2: Yes

5. Review Comments to the Author

Reviewer #1: The manuscript “Estimating and predicting the temporal information of apartment burglaries that possess imprecise timestamps: A comparative study using eight different temporal approximation methods in Vienna, Austria” introduced two novel temporal approximation methods and evaluated apartment burglaries in Vienna, Austria, for yearly and seasonal differences. The manuscript is a methodological study with an element of a case study. The analysis the paper promises is good, but the analysis delivered is less exciting. The methods are remarkably similar to existing studies. Therefore, I suggest emphasizing and in-depth analysis of seasonal differences.

1. Both the RWA, RWAext novel methods are similar to the methods provided in another published article (doi:10.3390/ijgi9060386) . Assumptions are also similar. So the methods are not very novelty. The difference between the novel methods in this paper and the existing methods should be identified.

The retrospectively weighted aoristic (RWA) temporal approximation method proposed in this manuscript is indeed novel and has not been mentioned in “doi:10.3390/ijgi9060386”. The retrospectively weighted aoristicext (RWAext) method is relatively new and is the same as the RTAext method in “doi:10.3390/ijgi9060386”.

We have clarified the difference between the novel methods in this manuscript and the existing novel methods in “doi:10.3390/ijgi9060386” throughout the revised manuscript, including the Abstract, 1. Introduction, 3.1. Research Objectives, 3.3.1. Temporal Approximation Methods, 4.4. Limitations, and 5. Conclusion.

2. Line 42: Please also include other keywords to mention the temporal approximation methods you introduced.

“Retrospectively weighted aoristic” and“retrospectively weighted aoristicext” methods are added to keywords

3. Line 269: As the Euqation (1) you mentioned shows the principle of the RWA method, please also use an Equation to show the principle of the RWAext method.

The equation of the RWAext method is added to Subsection 3.3.1. Temporal Approximation Methods as Equation (2) together with a fictive example.

4. Line 278: It seems that the number of Section “3.3.1 Statistical Evaluation” was wrong.

Corrected

5. Line 336: It will be better to additionally compare approximated distributions of apartment burglaries with unknown times from 2013 to 2015 to apartment burglaries with known offense times from 2013 to 2015. I leave the final decision to the authors.

This research has two goals, first to approximate offense times of retrospective (historic) crimes that possess imprecise temporal information and second to evaluate approximated offense times of future crimes with imprecise temporal information. While the suggestion by this reviewer can be used to evaluate approximation methods of offense times for retrospective (historic) crimes with imprecise timestamps, it would not be able to evaluate approximated offense times of future crimes with imprecise temporal information. Hence this approach was not applied in this research.

6. Line 450: Authors should add other similar work to improve the scientific strength of the work. There are no references published in the past two years in the references section.

Three articles published in 2020 were added.

7. Line 519: Incomplete legend has been provided in Figure 1. And the meaning of vertical axis is not clear.

Complete legend has been provided and vertical axis has been labeled.

Start here: Reviewer #2: This study addresses the question of how to handle temporal uncertainty in the time coding of crime incidents. It compares 8 different strategies for addressing such uncertainty, and compares the results. It uses data from Vienna for these comparisons.

I like the general idea of the research question, as it is certainly an important one. It seems like the data are useful for addressing the question, although there were a few details I was unclear on.

It is useful to compare strategies to the two aoristic strategies that are included, given that they are often used in studies. I do think these are quite naïve strategies, particularly the initial aortic one that does not account for partial hours, so it might be useful to at least point out a few of the potential limitations we would expect from these strategies based on their assumptions.

A paragraph discussing the main limitation of both aoristic methods has been added to Section 1. Theoretical Background.

I found myself somewhat confused about what exactly the data provides. Is there different time information for crime incidents in different years? At points, it seemed like there was, and that fed into the strategy used. But at other points it seemed like there was not. It would be good to add a paragraph that makes it explicit what temporal information is available in which years. And which years have the “gold standard” of correct time, and how exactly is that “gold standard” determined? This should be presented in a very clear fashion, as currently the information about this seems to be spread at various points in the data section, and it was not very clear at points. This may mostly be a re-ordering of information in the data section.

More text was added to the third paragraph under Subsection 3.2. Data Preparation to make it very clear which three datasets were collected and applied in this research. Also, the two terms “precisely known time stamps” and “imprecisely known time stamps” are now used throughout the manuscript (read our answer to the following question next).

It was not clear to me at points whether the author(s) were referring to incidents with exact time of incident, incidents with a range of time period, or incidents with no time information (I’m not sure how the latter would occur; it seems like this should just have a very wide time range). But making this clearer would help the readability at points.

Two types of time stamps are used for incidents in this research. First, incidents with precisely (accurately) known time stamps (to the minute), and second, incidents with imprecisely (inaccurately) known time stamps not exceeding a 24-hour time period. Incidents with no time information are not included. The term “precisely known offense time” (or similar) is now used for the first type of time stamps and “imprecisely known offense time” (or similar) for the second type of time stamps throughout this manuscript.

I wondered if another strategy would be to use time from the beginning of the unknown time period? If a house was being cased, it might be that it would be burglarized shortly after the people left, regardless of what time that is. Just a thought.

This idea has been implemented in this study with one of the four naïve methods (tstart) and interestingly performed the poorest of all eight temporal approximation methods (compare results in Tables 2 and 3).

On page 15, it was not clear to me whether the correlation reported is based on specific cases, or the distribution? I believe it’s the latter, but that should be clarified.

Correlation statistics are based on the distribution.

These are some interesting findings. It would be useful to talk about the potential generalizability of them. There are many possible issues. How generalizable are the temporal patterns of crime in this city? How generalizable is this particular crime type to others? It would be good to explicitly talk about these issues in the discussion section, as they would be important to potential users of the technique, especially if they did not have the type of data used here to calibrate the model. They would need to assume some of the patterns in the crime data here, so how reasonable would that be?

A new paragraph (now paragraph 3) is added to Section 5. Conclusion to discuss these issues.

6. PLOS authors have the option to publish the peer review history of their article (what does this mean?). If published, this will include your full peer review and any attached files.

Do you want your identity to be public for this peer review? For information about this choice, including consent withdrawal, please see our Privacy Policy.

Reviewer #1: No

Reviewer #2: No

---

## [Decision Letter · Decision Letter 1]

9 Jun 2021

Estimating and predicting the temporal information of apartment burglaries that possess imprecise time stamps: A comparative study using eight different temporal approximation methods in Vienna, Austria

PONE-D-20-32940R1

Dear Dr. Leitner,

We’re pleased to inform you that your manuscript has been judged scientifically suitable for publication and will be formally accepted for publication once it meets all outstanding technical requirements.

Kind regards,

Chaowei Yang

Academic Editor

PLOS ONE

Additional Editor Comments (optional):

Please make minor adjustment to your manuscript as noted by one of the reviewers and submitted when a final email is sent from PLOS One. 

Reviewers' comments:

Reviewer's Responses to Questions

**Comments to the Author**

1. If the authors have adequately addressed your comments raised in a previous round of review and you feel that this manuscript is now acceptable for publication, you may indicate that here to bypass the “Comments to the Author” section, enter your conflict of interest statement in the “Confidential to Editor” section, and submit your "Accept" recommendation.

Reviewer #1: All comments have been addressed

Reviewer #2: All comments have been addressed

2. Is the manuscript technically sound, and do the data support the conclusions?

Reviewer #1: Yes

Reviewer #2: Yes

3. Has the statistical analysis been performed appropriately and rigorously? 

Reviewer #1: Yes

Reviewer #2: Yes

4. Have the authors made all data underlying the findings in their manuscript fully available?

Reviewer #1: Yes

Reviewer #2: Yes

5. Is the manuscript presented in an intelligible fashion and written in standard English?

Reviewer #1: Yes

Reviewer #2: Yes

6. Review Comments to the Author

Reviewer #1: (No Response)

Reviewer #2: I appreciate the changes the authors have made to the manuscript, which have answered my questions and clarified features of the data and analyses. I just have a couple small suggestions remaining:

On page 16 near the top, it says that the Spearman’s rho of nearly .8 indicates “not systematically different”. I’m not sure what this term means---they are certainly similar, and I’d be inclined to just point that out.

A point to highlight in the discussion just a bit more is that you are uncertain if these crimes with exact times are a random sample of all crimes. This is unknowable, as you point out. But the Spearman rho of .954 does seem quite high, does it not? Maybe worringly so? I’d just add a couple sentences at the end acknowledging this a bit more.

7. PLOS authors have the option to publish the peer review history of their article (what does this mean?). If published, this will include your full peer review and any attached files.

Reviewer #1: No

Reviewer #2: No

---

## [Editor Report · Acceptance letter]

9 Aug 2021

PONE-D-20-32940R1 

Estimating and predicting the temporal information of apartment burglaries that possess imprecise time stamps: A comparative study using eight different temporal approximation methods in Vienna, Austria 

Dear Dr. Leitner:

I'm pleased to inform you that your manuscript has been deemed suitable for publication in PLOS ONE. Congratulations! Your manuscript is now with our production department. 

Kind regards, 

on behalf of

Dr. Chaowei Yang 

Academic Editor

PLOS ONE